# Mercury and Alzheimer’s Disease: Hg(II) Ions Display Specific Binding to the Amyloid-β Peptide and Hinder Its Fibrillization

**DOI:** 10.3390/biom10010044

**Published:** 2019-12-27

**Authors:** Cecilia Wallin, Merlin Friedemann, Sabrina B. Sholts, Andra Noormägi, Teodor Svantesson, Jüri Jarvet, Per M. Roos, Peep Palumaa, Astrid Gräslund, Sebastian K. T. S. Wärmländer

**Affiliations:** 1Department of Biochemistry and Biophysics, Stockholm University, 10691 Stockholm, Sweden; cecilia.wallin@dbb.su.se (C.W.); tesvant@gmail.com (T.S.); jyri.jarvet@dbb.su.se (J.J.); astrid@dbb.su.se (A.G.); 2Department of Chemistry and Biotechnology, Tallinn University of Technology, 19086 Tallinn, Estonia; merfrie@gmail.com (M.F.); andra.noormagi@gmail.com (A.N.); peep.palumaa@taltech.ee (P.P.); 3Department of Anthropology, National Museum of Natural History, Smithsonian Institution, Washington, DC 20560, USA; SholtsS@si.edu; 4The National Institute of Chemical Physics and Biophysics, 12618 Tallinn, Estonia; 5Institute of Environmental Medicine, Karolinska Institutet, 16765 Stockholm, Sweden; per.roos@ki.se; 6Department of Clinical Physiology, Capio St. Göran Hospital, 11219 Stockholm, Sweden

**Keywords:** mercury, Alzheimer’s disease, amyloid aggregation, metal–protein binding, neurodegeneration

## Abstract

Brains and blood of Alzheimer’s disease (AD) patients have shown elevated mercury concentrations, but potential involvement of mercury exposure in AD pathogenesis has not been studied at the molecular level. The pathological hallmark of AD brains is deposition of amyloid plaques, consisting mainly of amyloid-β (Aβ) peptides aggregated into amyloid fibrils. Aβ peptide fibrillization is known to be modulated by metal ions such as Cu(II) and Zn(II). Here, we study in vitro the interactions between Aβ peptides and Hg(II) ions by multiple biophysical techniques. Fluorescence spectroscopy and atomic force microscopy (AFM) show that Hg(II) ions have a concentration-dependent inhibiting effect on Aβ fibrillization: at a 1:1 Aβ·Hg(II) ratio only non-fibrillar Aβ aggregates are formed. NMR spectroscopy shows that Hg(II) ions interact with the N-terminal region of Aβ(1–40) with a micromolar affinity, likely via a binding mode similar to that for Cu(II) and Zn(II) ions, i.e., mainly via the histidine residues His6, His13, and His14. Thus, together with Cu(II), Fe(II), Mn(II), Pb(IV), and Zn(II) ions, Hg(II) belongs to a family of metal ions that display residue-specific binding interactions with Aβ peptides and modulate their aggregation processes.

## 1. Introduction

Alzheimer’s disease (AD) is a progressive, irreversible, and currently incurable neurodegenerative disorder, and the leading cause of dementia worldwide [1,2]. Identifying molecular targets [3] and/or modifiable risk factors related to disease onset and/or early progression is imperative [4,5]. AD risk factors so far identified include advanced age [6,7], genetic mutations associated especially with the *AβPP* and *ApoE* genes [8,9,10,11,12,13,14], life style [15,16,17], air pollution including tobacco smoking [18,19,20,21,22], cardio-vascular diseases [23], diabetes [24], traumatic brain injury [25,26], and metal exposure [27,28,29].

The major characteristic AD lesion in the brain is the presence of extracellular amyloid plaques, consisting mainly of amyloid-β (Aβ) peptides aggregated into insoluble fibrils [30] that display the cross-β secondary structure common for many amyloid fibrils [31,32]. The Aβ peptides, produced by two-step enzymatic cleavage of the membrane-bound amyloid-β precursor protein (AβPP), comprise 37–43 residues and are intrinsically disordered in aqueous solution. Aβ peptides have limited solubility in water, as the central and C-terminal Aβ segments are hydrophobic and may fold into a hairpin conformation upon aggregation [33]. The charged N-terminal segment is, however, hydrophilic and interacts readily with cationic molecules and metal ions [34,35,36,37,38]. The Aβ fibrils and plaques are the end-product of an Aβ aggregation process [37,39,40] involving extra- and/or intracellular formation of intermediate, soluble, and likely neurotoxic Aβ oligomers [41,42,43,44] that may transfer from neuron to neuron via exosomes [45,46]. Aβ_42_ oligomers appear to be the most cell-toxic [43], and oligomer formation appears to be influenced by both hydrophobic and electrostatic effects originating from interactions with, e.g., cellular membranes, metal ions, small molecules, and other proteins [24,37,47,48,49,50,51,52,53,54,55,56].

A second pathological lesion in AD brains is the presence of intracellular neurofibrillary tangles composed of aggregated hyperphosphorylated tau proteins [57,58]. Currently, the most plausible connection between AD and aggregates of Aβ and tau has been formulated in the so-called amyloid cascade hypothesis, but the underlying causative links are not fully understood [39,53,59]. AD brains, furthermore, typically display signs of neuroinflammation [60], increased levels of free oxygen radicals [61,62], and altered concentrations of different metal ions indicative of metal dyshomeostasis [63,64].

The proposed involvement of metal ions in AD pathology [65,66,67] is supported by a number of observations. Accumulation of metal ions such as those of Ca, Cu, Fe, and Zn has been found in AD plaques [68,69,70] and in phosphorylated tau tangles [71]. Cu(II), Fe(II), Mn(II), Pb(IV), and Zn(II) ions are known to bind to specific residues in the Aβ peptides and modulate their aggregation [22,38,65,72,73]. The metal ions are typically coordinated to one or more of the three histidines—His6, His13, and His14—and possibly also to other N-terminal residues such as the negatively charged Asp1 and Glu11 [73]. The metal–peptide interactions are transient, with binding affinities in the nanomolar or micromolar range [74,75,76], allowing formation of dynamic equilibria between different binding modes. The Aβ precursor protein AβPP also binds copper and zinc ions [77], and part of its physiological role might be connected with the regulation of Cu(II) and Zn(II) concentrations in the neuronal synapses, where these ions are released into the synaptic clefts [66] and where Aβ aggregation may be initiated [78]. Elevated Aβ concentrations have been observed in animals and cells exposed to metals such as Ag, As, Cd, Cu, Mn, Pb, and Hg [79,80,81,82,83,84,85,86,87,88,89].

A possible role of Hg in AD pathology is intriguing but also controversial [90]. Increased Hg levels were reported in early studies of brains [91] and blood [92,93] of AD patients [94]. Some later studies have, however, failed to confirm these observations [90,94,95,96]. While Hg is a well-known neurotoxicant, its toxic mechanisms are not fully understood at the molecular level [88,94,97,98,99], and metallic, inorganic, and organic Hg compounds are known to have different toxicity mechanisms and exposure pathways [94,97,98,100].

Few studies have so far investigated the possible influence of Hg(II) and other non-essential metal ions on the molecular events that occur in neurodegenerative diseases, such as aggregation of the Aβ peptide in AD [20,22,29,87,101]. Although Hg(II) ions are known to affect protein aggregation and misfolding in general [102,103,104], we are only aware of one study investigating possible molecular interactions between Hg(II) ions and Aβ peptides [105].

Here, we use solution nuclear magnetic resonance (NMR) and fluorescence spectroscopy, together with solid-state atomic force microscopy (AFM), to study the binding interactions between inorganic Hg(II) ions and Aβ_40_ and Aβ_42_ monomers, and the effects of Hg(II) ions on the Aβ amyloid aggregation process and fibril formation from a biophysical perspective.

## 2. Materials and Methods

### 2.1. Sample Preparation

Recombinant Aβ_42_ peptides, with the primary sequence DAEFR_5_HDSGY_10_EVHHQ_15_KLVFF_20_AEDVG_25_SNKGA_30_IIGLM_35_VGGVV_40_IA, were purchased lyophilized from rPeptide (Bogart, GA, USA), together with shorter recombinant Aβ_16_ peptides comprising the first 16 residues in the above sequence. The Aβ_16_ and Aβ_42_ peptides were dissolved in 100% hexafluoroisopropanol (HFIP) at concentrations of 100 µM to disassemble preformed peptide aggregates, and then aliquoted into smaller samples of 10–100 µg. The HFIP was evaporated in vacuum and the resulting Aβ films were stored at −80 °C. The peptide concentrations were determined by weight. For the experiments with the Aβ_16_ and Aβ_42_ peptides, Hg(II) solutions derived from a Hg(NO_3_)_2_ salt were used.

Recombinant unlabeled or uniformly ^15^N- or ^13^C–^15^N-labeled Aβ_40_ peptides, with a sequence identical to that of Aβ_42_ (above) but lacking the last two residues, were bought lyophilized from AlexoTech AB (Umeå, Sweden). The peptides were stored at −80 °C until used. The peptide concentration was determined by weight, and the peptide samples were dissolved to monomeric form immediately before each measurement. In brief, the peptides were dissolved in 10 mM sodium hydroxide, pH 12, at a 1 mg/mL concentration, and sonicated in an ice-bath for at least 3 min to avoid having pre-formed aggregates in the peptide solutions. The peptide solution was further diluted in 20 mM sodium phosphate buffer, and all sample preparation steps were performed on ice. For the experiments with the Aβ_40_ samples, Hg(II) solutions derived from a HgCl_2_ salt were used.

### 2.2. Fluorescence Spectroscopy

#### 2.2.1. ThT Fluorescence as a Probe for Aβ Aggregation Kinetics

To monitor the Aβ fibrillization kinetics, 15 µM monomeric Aβ_40_ peptides were incubated in 20 mM sodium phosphate buffer pH 7.35 at +37 °C in the presence of varying concentrations of HgCl_2_ (0, 0.2, 0.8, 1.5, 3.0, and 15 µM) and 40 μM Thioflavin T (ThT). ThT is a benzothiazole dye that displays increased fluorescence intensity when bound to amyloid material [106]. The ThT dye was excited at 440 nm, and the fluorescence emission at 480 nm was measured every 3 min in a 96-well plate in a FLUOstar Omega microplate reader (BMG LABTECH, Germany). The Aβ_40_ samples were incubated under quiescent conditions for two days at +37 °C, and four replicates a la 100 µL per condition were measured. The assay was repeated at least three times. The Aβ aggregation kinetic parameters τ_½_ and r_max_ were calculated by sigmoidal curve fitting according to Equation (1) [107] using four replicates per condition:(1)$$F\left(t\right)=F_{0}+\frac{A}{1+\exp\left[r_{\max}\left(\tau_{1/2}-t\right)\right]}$$
where F_0_ is the fluorescence intensity baseline, A is the fluorescence intensity amplitude, r_max_ is the maximum growth rate, and τ_½_ is the time when half the Aβ monomer population has been depleted. The aggregation lag time was defined by $\tau_{\mathrm{lag}}=\tau_{½}-\frac{2}{r_{\max}}$.

The fibrillization kinetics of Aβ_42_ was studied using HFIP-treated Aβ_42_ dissolved at a concentration of 20 µM in 0.02% NH_3_, and then diluted to a final concentration of 5 µM in a buffered solution containing 20 mM HEPES pH 7.3, 100 mM NaCl, and 5 µM ThT. Aβ_42_ samples of 500 µL were prepared with different Hg(NO_3_)_2_ concentrations (i.e., 0, 5, 7.5, 10, and 15 µM) in a quartz cuvette with 0.5 cm path length, and incubated at +45 °C with high-speed magnetic stirring. The ThT fluorescence emission at 480 nm was measured over time on an LS-55 fluorescence spectrophotometer (PerkinElmer Inc., Waltham, MA, USA) using an excitation wavelength of 440 nm.

#### 2.2.2. Tyrosine Fluorescence Quenching Reporting on Hg(II)·Aβ Binding Affinity

The binding affinity between Aβ_16_ peptides and Hg(II) ions was evaluated from Cu(II)·Hg(II) binding competition experiments [108]. The Cu(II)·Aβ_16_ affinity was measured via the quenching effect of Cu(II) ions on the intrinsic fluorescence of Tyr10, the only fluorophore in native Aβ peptides. HFIP-treated Aβ_16_ was dissolved in 10 mM NaOH at a concentration of 80 µM and divided into 60 µL aliquots that were stored at −20 °C, and then used within the same day to prepare samples containing 8 µM Aβ_16_ in 100 mM NaCl and 20 mM HEPES pH 7.4, or in 20 mM MES pH 5.0. Titrations with increasing concentrations of CuCl_2_ were conducted at +25 °C using samples with or without Hg(NO_3_)_2_ added in different concentrations, i.e., 50 and 100 µM Hg(II) at both pH 5.0 and pH 7.4. The fluorescence emission intensity at 305 nm (excitation wavelength 270 nm) was recorded using an LS-55 fluorescence spectrophotometer (Perkin-Elmer Inc., Waltham, MA, USA) equipped with a magnetic stirrer. The titrations were carried out by consecutive additions of 1 µL aliquots of 0.3–12 mM stock solutions of CuCl_2_ to 600 µL Aβ_16_ peptide solutions in quartz cuvettes with 5 mm path length. After each addition of CuCl_2_, the solution was stirred for 30 s before recording fluorescence emission spectra.

From the Cu(II) titration curves, recorded in the absence or presence of Hg(II) ions, the apparent dissociation constant (K_D_^app^) of the Cu(II)·Aβ_16_ complex was determined as the concentration of Cu(II) ions that reduced the Tyr10 fluorescence by 50% (as Hg(II) ions alone have no have no significant effect on the Tyr10 fluorescence intensity: data not shown). The Cu(II)·Aβ_16_ K_D_^app^ values were then plotted versus the concentration of Hg(II) ions present in the samples, and the linear dependence was extrapolated to the point where the K_D_^app^ value at zero Hg(II) had been increased by a factor of 2. Assuming competitive binding between Hg(II) and Cu(II) ions to the same Aβ_16_ binding site, the Hg(II) concentration at this point is approximately equal to the apparent dissociation constant (K_D_) for the Hg(II)·Aβ_16_ complex.

### 2.3. AFM Imaging

Solid-state AFM images were recorded using a ScanAsyst unit (Bruker Corp., Billerica, MA, USA) operating in peak-force mode in air with a sample rate of 1.95 Hz and a resolution of 512 × 512 pixels. At the end of the fluorescence spectroscopy ThT kinetic experiments with Aβ_40_ samples (above), the aggregated Aβ end products were diluted with distilled water (5 μL sample in 100 μL distilled water) and applied on freshly cleaved mica substrates. After 20 min incubation, the mica substrates were washed three times with distilled water and left to air-dry. This protocol was optimized to obtain images of Aβ fibrils, and might not be optimal for other Aβ aggregates or Hg(II)·Aβ complexes. 

### 2.4. NMR Spectroscopy

Bruker Avance 500 and 700 MHz spectrometers equipped with cryogenic probes were used to record two-dimensional (2D) ^1^H–^15^N-HSQC (heteronuclear single quantum coherence) and 2D ^1^H–^13^C-HSQC spectra at +5 °C or +25 °C of 84 μM monomeric Aβ(1–40) peptides (either ^15^N-labeled or ^13^C-^15^N-labeled) in 20 mM sodium phosphate buffer at pH 7.35 (90/10 H_2_O/D_2_O), either with or without 50 mM sodium dodecyl sulfate (SDS) detergent, and before and after titration with HgCl_2_. Because NMR spectroscopy is a non-sensitive high-resolution method that requires high protein concentration and pure samples, the chosen Aβ_40_ concentration was 84 μM. The Aβ_40_ NMR sample was stable during the experimental time required for the HgCl_2_ titration series. Excess concentration of SDS (around 50 mM) well above the critical micelle concentration (CMC; around 8 mM) was used as a membrane-mimicking environment. SDS micelles are simple *in vitro* membrane models with a small size range suitable for NMR spectroscopy [109]. The experiments performed in buffer were recorded at +5 °C due to optimal signal intensity and to limit the peptide aggregation during the experimental time. A temperature of +25 °C was used for the experiments with SDS to avoid precipitation of the SDS detergent. The NMR data were processed with the Topspin version 3.2 software and chemical shifts were referenced to the ^1^H signal of trimethylsilylpropanoic acid (TSP). The Aβ_40_ HSQC crosspeak assignment in buffer [110,111,112] and in SDS micelles [113] is known from previous work. Chemical shift differences between the spectra of Aβ_40_ with and without Hg(II) ions were calculated as the standard weighted average, i.e., Δδ = (((Δδ_N_/5)^2^ + (Δδ_H_)^2^)/2)^1/2^ [114,115].

The intensity decrease of the 2D ^1^H–^15^N-HSQC crosspeaks is likely caused by two effects, both related to the presence of Hg(II) ions. The first effect is Hg(II)-induced non-amyloid aggregation of Aβ peptides (i.e., precipitation or formation of non-fibrillar amorphous aggregates), which reduces the concentration of Aβ monomers and thus the intensities of all Aβ NMR signals. The second effect decreases more selectively the intensities of the (N-terminal) NMR crosspeaks affected by Hg(II) ion binding, and appears to be caused by chemical exchange on an intermediate time scale. Thus, this second effect reports on the Hg(II)·Aβ binding affinity. To compensate for the first effect, the N-terminal crosspeak intensities were normalized by dividing their signal intensities with the corresponding intensities of the C-terminal crosspeaks, which are affected only by the loss of monomer concentrations. The relative N-terminal intensities reflect only the chemical exchange effect, and were fitted to Equation (2) [116] to yield an estimated apparent dissociation constant (K_D_^app^) for the Hg(II)·Aβ_40_ complex:(2)$$I=I_{0}+\frac{I_{\infty }-I_{0}}{2\cdot \left[A\beta \right]}\cdot \left(\left(K_{D}+\left[\mathrm{Hg}\right]+\left[A\beta \right]\right)-\sqrt{{\left(K_{D}+\left[\mathrm{Hg}\right]+\left[A\beta \right]\right)}^{2}-4\cdot \left[\mathrm{Hg}\right]\cdot \left[A\beta \right]}\right)$$
where I_0_ is the initial fluorescence intensity without Hg(II) ions, I_∞_ is the steady-state (saturated) intensity at the end of the titration series, [Aβ] is the peptide concentration, [Hg] is the concentration of added Hg(II) ions, and K_D_ is the dissociation constant of the Hg(II)·Aβ complex. The model assumes a single binding site. As no corrections for buffer conditions were made, the calculated dissociation constant should be considered to be apparent.

## 3. Results

### 3.1. ThT Fluorescence: Kinetic Effects of Hg(II) Ions on the Aβ_40_ and Aβ_42_ Aggregation Processes

The fluorescence intensity of the amyloid-marker molecule ThT was measured when 15 µM Aβ_40_ samples were incubated for two days, together with different concentrations of HgCl_2_ during quiescent conditions (Figure 1 and Appendix A). Clear concentration-dependent effects of Hg(II) on the Aβ_40_ aggregation kinetics were observed (Figure 1). The increased noise level in the kinetic curves following the elongation phase might originate from light scattering effects from large and heterogeneous aggregates. Fitting Equation (1) to the ThT fluorescence curves yielded the kinetic parameters τ_lag_, τ_½_, r_max_, and ThT end-point fluorescence intensity (Figure 1; Table 1). For 15 µM Aβ_40_ alone, the aggregation lag time (τ_lag_) was approximately 7 h under our experimental conditions, and the aggregation halftime (τ_½_) was 10 h (Table 1). These kinetic parameters were not much affected by addition of 0.8 µM Hg(II), but clearly increased when 15 µM Aβ_40_ was incubated in the presence of 1.5 µM or 3 µM Hg(II) (Figure 1). In the presence of 3 µM Hg(II), τ_lag_ was around 13 h and τ_½_ was around 20 h (Table 1). When 15 µM Hg(II) was added, no increase in ThT fluorescence intensity was observed during the entire incubation period, indicating that ThT-active amyloid aggregates did not form. This is consistent with the end-point ThT fluorescence intensity levels, which reflect the amount of ThT-active aggregates at the end of the incubation. These end-point ThT levels strictly decrease with increasing Hg(II) concentrations (Table 1). The maximum amyloid growth rate, r_max_, displayed more variation: It first increased from 0.6 to around 0.8 when small amounts (0.8 and 1.5 µM) of Hg(II) were added, and then decreased to 0.3 in the presence of 3 µM Hg(II) ions (Figure 1; Table 1).

For the Aβ_42_ peptide incubated under agitation conditions, the ThT kinetics curves again showed a clear dependence on the Hg(II) concentration. Increasing concentrations of Hg(II) ions induce strictly increasing aggregation half times and lag times, and strictly decreasing maximum growth rates and ThT end-point intensities (Appendix A). Although Aβ_42_ peptides are more prone to aggregate than Aβ_40_ peptides, similar trends for the effects of Hg(II) ions on Aβ aggregation kinetics were observed for the two peptide versions. Thus, these concentration-dependent effects do not appear to depend on the particular experimental conditions.

### 3.2. AFM Imaging: Effects of Hg(II) Ions on Aβ_40_ Aggregate Morphology

AFM images (Figure 2 and Appendix A) were recorded for the aggregation products present at the end of the ThT kinetics experiments. First, 15 µM Aβ_40_ alone formed typical amyloid fibrils with a height of about 10 nm (Figure 2A and Appendix A), which is a typical size for Aβ fibrils formed in vitro [42,117]. The aggregates of 15 µM Aβ_40_ formed in the presence of low concentrations (0.8, 1.5, and 3.0 µM) of HgCl_2_ displayed shorter fibril-like structures (Figure 2B–D and Appendix A). No amyloid aggregates were observed when the Aβ_40_ sample was incubated together with equimolar amounts of HgCl_2_ (Figure 2E and Appendix A). These results are consistent with the concentration-dependent inhibitory effect of Hg(II) ions on Aβ fibrillization observed with the ThT assays (Figure 1).

### 3.3. NMR Spectroscopy: Molecular Interactions Between Hg(II) Ions and Aβ_40_ Monomers

High-resolution NMR experiments were conducted to investigate if residue-specific molecular interactions could be observed between Hg(II) ions and monomeric Aβ_40_ peptides (Figure 3 and Figure 4, Appendix A). Two-dimensional ^1^H–^15^N-HSQC spectra showing the amide crosspeak region for 84 µM monomeric ^13^C–^15^N-labeled Aβ_40_ peptides are presented in Figure 3, recorded either without (Figure 3A) or with 50 mM SDS detergent (Figure 3C), before and after addition of 80 μM Hg(II) ions (Figure 3A) or 30 μM Hg(II) ions (Figure 3C). Addition of Hg(II) ions selectively induces loss of signal intensity mainly for amide crosspeaks corresponding to N-terminal Aβ_40_ residues (Figure 3A,B), indicating selective Hg(II) binding in this region. These effects are clearly concentration-dependent, as seen in Appendix A.

Figure 4 shows 2D NMR ^1^H–^13^C-HSQC spectra for aromatic (Figure 4A) and C_α_–H (Figure 4C) crosspeaks of 84 μM unstructured ^13^C-^15^N–Aβ_40_ peptides in phosphate buffer, before and after addition of 80 μM Hg(II) ions. Again, crosspeaks corresponding to N-terminal Aβ_40_ residues display the largest intensity loss (Figure 4B,D). Although these NMR observations give no direct information about the metal-binding coordination, the observed loss of NMR signal for specific residues is likely caused by intermediate chemical exchange on the NMR time-scale between a free and a metal-bound state of the Aβ_40_ peptide, similar to the effect induced by Zn(II) ions [118]. The histidine residues His6, His13, and His14 are markedly affected by the Hg(II) ions (Figure 4B), and might be the main metal-binding ligands. The general loss of Aβ_40_ crosspeak signal intensity induced by addition of HgCl_2_ (Figure 3B, Figure 4 and Appendix A) indicates that Hg(II) ions promote formation of large Aβ aggregates, where some of them are too large to be observed with HSQC NMR and some simply precipitate out of the solution. Thus, the NMR data presented in Figure 3 and Figure 4 mainly stem from interactions between Hg(II) ions and non-aggregated Aβ_40_ monomers. 

Specific binding of Hg(II) ions to the Aβ N-terminal region is observed also when the Aβ_40_ peptides are bound to SDS micelles (Figure 3C,D), used here as a simple bio-membrane model [109,119]. The ^1^H–^15^N-HSQC spectrum for Aβ_40_ in SDS micelles (Figure 3C) corresponds to a partly α-helical Aβ conformation, where the central (residues 15–24) and C-terminal (residues 29–35) Aβ segments are inserted as α-helices into the micelles [113]. The N-terminal segment is unstructured and located outside the micelle surface, where it can interact with binding agents such as metal ions [116,120]. Addition of Hg(II) ions induces a concentration-dependent intensity loss for N-terminal Aβ_40_ amide crosspeaks (Figure 3D and Appendix A), but there is no general loss of amide crosspeak intensity as the Aβ peptides do not aggregate when bound to SDS micelles (at least for Aβ/micelle ratios < 1; cf. Figure 3B,D). Addition of Hg(II) ions induces small changes in the positions of some Aβ_40_ amide crosspeaks (Figure 3C,G), indicating a structural reorganization of the Aβ_40_ peptides inside the micelles. This effect might be related to the previously observed increase in helix supercoiling of SDS-bound Aβ induced by Cu(II) ions [116], even though the chemical shifts induced by the Cu(II) and Hg(II) ions are slightly different.

The relative intensities of the Aβ_40_
^1^H–^15^N-HSQC NMR amide crosspeaks corresponding to N-terminal residues were corrected for the general loss of signal intensity caused by Hg(II)-induced aggregation and plotted as a function of added Hg(II) ions (Figure 3E). Fitting Equation (2) globally to all of the curves in Figure 3E produced an apparent Hg(II)·Aβ_40_ K_D_ value of 11 ± 4 μM.

### 3.4. Fluorescence Spectroscopy: pH-Dependence of the Hg(II) Binding Affinity to the Aβ_16_ Monomer

The Aβ_16_ peptide was used to investigate the pH-dependence of the Hg(II)·Aβ binding affinity, as it is less prone to aggregate during measurements than full-length Aβ peptides. Because Cu(II) ions quench the intrinsic fluorescence of tyrosine residues, while Hg(II) ions do not, the Hg(II)·Aβ_16_ binding affinity was measured via Cu(II)·Hg(II) competition experiments. Figure 5A,B shows fluorescence quenching data for the Tyr10 residue of 8 µM Aβ_16_, obtained via stepwise additions of Cu(II) ions in the absence and presence of different concentrations of Hg(II) ions at either pH 7.4 or pH 5.0. The K_D_^app^ value for the Cu(II)·Aβ_16_ complex at pH 7.4 was estimated to be approximately 22 µM in the absence of Hg(II) ions. This value is somewhat higher than the previously reported K_D_^app^ values of around 0.5–3 µM, which might be due to different sample preparations and different experimental conditions during the measurements [74,75,76]. When increasing concentrations of Hg(II) ions are present in the pH 7.4 samples, the titration curves are shifted towards weaker Cu(II) affinities, showing that the Hg(II) and Cu(II) ions compete for binding to the Aβ_16_ peptide (Figure 5A). The K_D_^app^ values for the Cu(II)·Aβ_16_ complex, obtained from the Cu(II) titration series in the presence of 0, 50, and 100 µM Hg(II) ions, are, respectively, 22, 28, and 35 µM. Plotting these K_D_^app^ values vs. the Hg(II) concentration (Figure 5A, insert) produced a straight line that was extrapolated to the point where K_D_^app^ is two times the K_D_^app^ value in absence of Hg(II) ions. At this point, the Hg(II) concentration, which was found to be 170 µM, should be approximately equal to the apparent K_D_ for the Hg(II)·Aβ_16_ complex.

At pH 5.0, addition of up to 100 µM of Hg(II) ions has no influence on the Cu(II) titration curve (Figure 5B), demonstrating that at this lower pH, the binding of Hg(II) ions to Aβ_16_ is too weak to compete with the binding of Cu(II) ions. This weaker binding of Hg(II) ions to Aβ_16_ at lower pH is most likely related to protonation of the His residues [121,122,123].

## 4. Discussion

### 4.1. The In Vitro Analyses of the Hg(II)·Aβ Complexes and Their Aggregation

Our ThT fluorescence results and AFM images show that Hg(II) ions have a clear and concentration-dependent inhibitory effect on the fibrillization of both Aβ_40_ (Figure 1 and Figure 2) and Aβ_42_ (Appendix A) peptides. Instead of forming amyloid fibrils, the Aβ peptides appear to form non-fibrillar amorphous aggregates in the presence of Hg(II) ions. According to the NMR data (Figure 3 and Figure 4), this effect appears to be related to specific binding interactions between Hg(II) ions and the Aβ N-terminal residues. Typical metal-coordinating residues in proteins are cysteines, histidines, and the negatively charged aspartic and glutamic acids. Here, the NMR signals of the Aβ residues His6, His13, and His14 display the most pronounced signal attenuation when Hg(II) ions are added (Figure 4), indicating that the main binding ligands are these three histidines, which coordinate transition metal ions via their imidazole groups. This conclusion is supported by the Hg(II) ions being able to compete with Cu(II) ions for binding to Aβ_16_ at pH 7.4 (Figure 5A), indicating that similar binding ligands are involved. Cu(II) ions, as well as Fe(II), Mn(II), and Zn(II) ions, have previously been shown to bind to the His residues of Aβ peptides [38,72,75,124,125]. For Pb(IV) ions, both the Tyr10 and the His residues appear to be potential binding ligands [22]. Metal ions such as Ca(II), Cd(II), Cr(III), and Pb(II), on the other hand, do not display residue-specific binding to Aβ monomers [22,126]. The observed His-based binding of Hg(II) ions to Aβ is somewhat surprising, as Hg(II) ions are known to prefer binding ligands such as thiol (−SH) and selenohydryl (−SeH) groups [104,127]. Because Aβ peptides lack Cys and Sec (also known as Se–Cys) residues, it has earlier been suggested that Aβ will not bind Hg(II) ions [128].

The Aβ_16_, Aβ_40_, and Aβ_42_ peptides share the same first 16 residues, and we therefore expect these peptide versions to have similar N-terminal Hg(II) binding modes. The observed binding affinities are, however, not identical. The K_D_^app^ value obtained for Hg(II) binding to Aβ_40_ is 11 ± 4 μM, while for binding to Aβ_16_ it is around 170 μM. Furthermore, addition of 30 µM Hg(II) ions to the Aβ_40_ peptides in SDS micelles induces around 50% intensity loss of the N-terminal NMR crosspeak signals (Appendix A), suggesting an apparent Hg(II)·Aβ_40_ affinity around 30 µM under those sample conditions. These different results are likely caused by various factors known to influence the binding affinity, such as differences in peptide length, the Aβ aggregation state, and experimental conditions, including buffer type, ionic strength, and pH: As the histidine ligands are very sensitive to protonation around pH 7, small differences in the pH conditions may lead to noticeable differences in binding strength [108,122]. The Hg(II)·Aβ_16_ affinity measurements were conducted at +25 °C at high salt concentrations (20 mM HEPES pH 7.4 + 100 mM NaCl) on non-aggregated Aβ_16_ samples (as Aβ_16_ does not readily aggregate). The Hg(II) affinity measurements to Aβ_40_ in buffer were conducted at +5 °C at moderate salt concentrations (20 mM sodium phosphate buffer, pH 7.35), on samples that aggregated during the measurements due to the added Hg(II) ions (Figure 3B). The Aβ peptides bound to SDS micelles do not aggregate (at least for Aβ/SDS micelle ratios < 1), but even though previous studies have shown that Aβ peptides positioned in SDS micelles can bind Cu(II) and Zn(II) ions with similar binding affinity as in aqueous solution [116], the binding of Hg(II) ions to SDS-bound Aβ peptides may be somewhat affected by the negatively charged SDS micelles. Although all employed approaches to investigate the Hg(II)·Aβ binding affinity have potential disadvantages, the Cu(II)·Hg(II) competition studies for binding to Aβ_16_ constitute an indirect method based on the unverified assumption of a shared binding site. Because the NMR experiments measure direct Hg(II)·Aβ_40_ interactions, we consider them more reliable, and tentatively conclude that the Hg(II)·Aβ binding affinity is in the approximate range of 10–30 µM. Such a conclusion is compatible with the concentration-dependent effects of total fibril inhibition at equimolar concentrations, as observed in the ThT kinetic assay and AFM images (Figure 1 and Figure 2, Appendix A).

Thus, the apparent Hg(II)·Aβ_40_ dissociation constant seems to be in a similar range as the 30–60 µM range reported for the Mn(II)·Aβ_40_ complex [72] and the 1–100 µM range reported for the Zn(II)·Aβ_40_ complex [75,76,124], but weaker than the 0.5–10 µM range observed for the Cu(II)·Aβ_40_ complex [72,74,75,76,116,122,124]. When corrected for buffer effects, K_D_ values of 1–50 nM for Aβ·Cu(II) and 0.1–1 µM for Aβ·Zn(II) have been calculated [75,76].

Detailed analyses of Cu(II) and Zn(II) binding to Aβ have shown different effects on peptide aggregation depending on the experimental conditions [38,118,129,130]. The different aggregation pathways might arise from the coordination of multiple Aβ peptides to the same metal ion, which likely is an important factor in Aβ aggregation [130,131]. Although our NMR data indicate that the histidine residues are the main binding ligands to Hg(II) (Figure 4), other N-terminal residues such as Asp1, Glu3, Asp7, Tyr10, and Glu11 are also possible binding partners. When multiple Aβ peptides are involved in coordinating the same metal ion, a multitude of possible binding arrangements are possible, and they are expected to have different effects on the aggregation process. Factors such as pH and salt conditions will influence which binding arrangement is the most favorable, although dynamic equilibria between different arrangements are expected. Investigating such binding modes, and metal binding to Aβ oligomers in general, is an important task for future studies. When a single Aβ peptide coordinates a single metal ion, the peptide appears to adopt a structure unsuitable for fibril formation [118]. Coordinating one metal ion to two or more Aβ peptides usually promotes aggregation, but not necessarily fibrillization [130,131]. In particular, supra-stoichiometric amounts of metal ions often induce rapid formation of amorphous aggregates instead of fibrils [38,132]. Our current results (Figure 1 and Figure 2, Appendix A) show that Hg(II) ions affect Aβ aggregation in a similar fashion as Cu(II) and Zn(II) ions, and prevent fibrillization already at a 1:1 Hg(II)·Aβ ratio. This finding is in agreement with the hypothesis that heavy metals might have a general capacity to induce aggregation of unstructured peptides/proteins [103,104]. As our NMR data indicate that addition of Hg(II) promotes rather than inhibits Aβ aggregation (Figure 3, Appendix A), Hg(II) ions appear to direct the Aβ aggregation pathway towards unstructured (i.e., non-fibrillar) aggregates. Such unstructured aggregates of Hg(II)·Aβ complexes likely have different electrostatic and hydrophobic properties than Aβ fibrils, which might explain why they are not readily visible in our AFM images (Figure 2 and Appendix A): The incubated Aβ samples were deposited on mica plates using a protocol optimized for imaging of Aβ fibrils.

The Hg(II) ions bind also to Aβ peptides positioned in SDS micelles (Figure 3C,D,G). This is not surprising, as the N-terminal metal-binding Aβ segment is known to be located outside SDS micelles [113], which are considered simplistic membrane models [109]. These results therefore suggest that Hg(II) ions might bind Aβ peptides located in cellular membranes. Such binding could be of biological relevance, as membrane disruption by Aβ oligomers has been suggested to be a neurotoxic mechanism in AD [44,48,133]. It has recently been reported that Hg(II) ions can block membrane channels formed by Aβ_42_ oligomers [105], which is in line with an earlier observation that membrane leakage of Ca(II) ions induced by Aβ oligomers can be blocked by histidine-binding metal ions and small molecules [134]. Given the inhibitory effects of Hg(II) ions on Aβ fibrillization, it would be interesting to investigate if Hg(II) ions can also modulate the formation of toxic or membrane-disrupting Aβ oligomers.

### 4.2. Mercury and AD: Clinical Studies and Sources of Exposure

Mercury has, for many decades, been implicated as a risk factor for AD, as elevated Hg levels have been found in early studies of brain and blood of AD patients [10,91,92,93,135,136,137]. Later studies have, however, failed to confirm these higher Hg levels in AD patients [94,95,96]. Because AD patients display altered metal dyshomeostasis [64] that manifests in different ways, including altered plasma/CSF (cerebrospinal fluid) ratios for various metal ions including Hg(II), it has been suggested that AD pathology may involve a compromised blood–CSF barrier [93,137]. On the other hand, the brain metal chemistry is notoriously complex, and studies on beluga whales and humans indicate that Hg(II) and other metal ions accumulate into different brain regions also for non-AD individuals [138,139,140]. One study with a limited number of patients reported elevated Hg concentrations in the two brain regions nucleus basalis of Meynert (NBM) and amygdala [135]. NBM is the major source of cholinergic innervation of the neocortex, and neuronal loss in this region is a well-known pathological feature of both AD and Parkinson’s disease [141].

It is currently unclear if the metal dyshomeostasis observed in AD patients is a cause or effect of the disease progression. Some uncertainties are present in all studies on Hg brain concentrations, due to the difficulties involved in determining metal concentrations in tissue samples and body fluids in general [142], especially for samples stored for a long time in biobanks and, in particular, for ions of Hg [97,143]. Another source of error is the problematic accuracy of AD diagnoses, especially when conducted without reliable biomarkers such as identification of AD plaques via PET scanning or MRI [144,145]. Because symptoms of chronic Hg exposure include personality changes and memory loss, especially for elderly people, Hg poisoning could conceivably be misdiagnosed as AD [10,94,146]. Yet, meta studies support a possible connection between Hg and AD [94]. Studies correlating AD incidence to the number of dental amalgam fillings are often difficult to interpret due to numerous sources of error, and generally show no or little correlation [94]. This suggests that if Hg is a risk factor for AD, then dental amalgam is not the critical source of Hg exposure [147].

Dental amalgam fillings are, nevertheless, the main source of human exposure to metallic Hg [97], which is converted via metabolic oxidation to inorganic Hg(II) ions [148]. Humans are also exposed to fair amounts of lipophilic methyl mercury (MeHg), mainly produced from Hg(II) ions by aquatic microbes [149] and readily bioaccumulates in fish and other marine organisms [97,150]. Less common is exposure to other forms of mercury, such as inorganic Hg_2_(II) and Hg_3_(II) polycations and organometallic dimethyl–Hg and ethyl–Hg complexes [97]. During the last decades, industrial release of Hg into the environment has decreased in the Western world but increased in the developing world, mainly due to gold mining and coal burning [128,146,151]. Thus, it still remains relevant to investigate the possible connection between Hg and AD.

### 4.3. Biological Relevance and Other Molecular Effects on AD Involving Hg Ions

The Hg concentration in the human brain is around 10–50 ng/g (approximately 0.1–0.3 µM) [91,135]. This is significantly lower than the 1–100 µM concentrations of Hg(II) ions used in our in vitro studies. Due to the requirements of the employed spectroscopic techniques, the 10–100 µM Aβ concentrations used in our experiments are also higher than the picomolar–nanomolar Aβ levels typically observed in human brains [152]. Local Aβ concentrations in, e.g., cell membranes may, however, be higher. The higher Aβ concentrations used in our experiments should promote peptide aggregation, but the effect of Hg(II) ions may depend more on the Hg(II)/Aβ ratio than the absolute concentrations. The total inhibition of Aβ fibrillization at 1:1 Hg(II)/Aβ_40_ ratio (Figure 1 and Figure 2) shows that small amounts of mercury in a critical location can have a large impact on Aβ aggregation. As Hg poisoning correlates with a variety of adverse effects on developing neurites [153], neurotransmission [154], and cognitive function [97,155], the amount of Hg that enters the brain after exposure events clearly has biological impact [10,88,90,94]. While Hg(II) ions do not easily pass the blood–brain barrier (BBB), metallic vapor mercury does [100]. Thus, if metallic vapor mercury passes the BBB and then becomes oxidized to Hg(II), these mercuric ions will be trapped inside the brain. MeHg easily passes across the BBB and the placenta, either by itself or bound to the amino acid cysteine: Such a complex is misrecognized as methionine by transport proteins and therefore freely transported throughout the body [156]. It is currently unclear if Hg is differently deposited in healthy and in AD brains, and if there is co-localization of Hg(II) ions and Aβ peptides in some brain compartments.

Mercury could conceivably affect AD pathology without directly interacting with the Aβ peptides themselves [90,94], for example, via toxic molecular mimicry [157], by promoting the aggregation of the tau fragment R2 [101] and phosphorylation of the tau protein as observed in SHSY5Y neuroblastoma cells [87], or via interactions between other forms of Aβ and Hg than those studied here. MeHg and Hg(II) ions bind to and affect the functions of important intracellular biomolecules with essential thiol (−SH) and selenohydryl (−SeH) groups, such as cysteine, homocysteine, metallothioneins, selenoproteins, glutathione (GSH), tubulin, ion channel proteins, transporters, metabolic enzymes, and N-methyl-D-aspartate (NMDA) receptors, thereby influencing or even damaging various tissues including nerve cells [94,97,153,157,158]. Hg exposure furthermore increased the release of Aβ in SHSY5Y neuroblastoma cells [87], which may promote Aβ aggregation. In Wistar rats exposed to MeHg, the Aβ levels increased in the hippocampus but decreased in the CSF, likely due to impaired Aβ transport [159]. Increased production of AβPP and decreased production of the Aβ-clearing enzyme neprilysin was observed in PC12 cells exposed to Hg(II) and MeHg [89]. In neuroblastoma cells exposed to HgCl_2_, the increased release of Aβ could be reversed by addition of melatonin [87], which is known to chelate metal ions [160]. Notably, one study using primary endothelial cells from transgenic mice reported increased AβPP expression and sAPPβ secretion in the presence of oxygen radicals [161].

Antioxidants such as GSH and many selenoproteins are known to be blocked by Hg [162], and one important toxic mechanism of various forms of Hg is disruption of the molecular defense against reactive oxygen species (ROS), especially in the mitochondria [98]. This mechanism may be important for both AD and general Hg intoxication. An impaired ROS defense obviously allows for higher concentrations of oxygen radicals, which, as stated above, was reported to promote Aβ production [161]. However, AD pathology appears to involve also cellular and molecular damage directly caused by oxygen radicals, likely formed by Fenton-type reactions with redox-active metal ions such as Cu(II)/Cu(I) and Fe(III)/Fe(II) [61,163,164]. The Hg-induced disruption of the ROS defense [88] likely promotes the ROS-related component of AD pathology [94,98]. This might be particularly relevant for the mitochondria, as they are often affected by Hg exposure [165,166], and as mitochondrial dysfunction is commonly observed in AD brain neurons [62,167,168,169].

Interestingly, AD and Hg intoxication have a common risk factor in the gene encoding the apolipoprotein E (ApoE). Hg exposure typically produces worse outcomes in individuals with the ApoEε4 allele [170,171,172,173], and this allele is linked also to an increased probability of developing AD [8,9,10,11,14]. The underlying reasons for this similar risk factor are unclear, however, and a number of explanations have been proposed [8,174], including the possibility that the beneficial ApoE variants may interact with and promote clearance of the Aβ peptide [175] or Hg ions [9,11]. ApoEε4 proteins might bind metal ions such as Hg(II) less efficiently than other ApoE isoforms, as they have two Arg residues in positions where ApoEε2 proteins have two −SH-containing Cys residues [11]. ApoEε3 has one Arg and one Cys residue. Whether the negative effects of the ApoEε4 allele are in fact related to a possibly reduced capacity for binding and eliminating Hg ions remains to be investigated. Nonetheless, studies investigating the relationship between AD and Hg exposure should benefit from taking into account the ApoE genotype of the studied individuals [173,174,176].

## 5. Conclusions

Hg(II) ions display specific binding to the N-terminal part of the Aβ peptide, likely coordinated mainly via the Aβ residues His6, His13, and His14, with an apparent Hg(II)·Aβ_40_ binding affinity in the micromolar range. The Hg(II) ions inhibit Aβ_40_ and Aβ_42_ fibrillization in a concentration-dependent manner, and at a 1:1 Hg(II)/Aβ ratio only non-fibrillar Aβ aggregates are formed. The observed molecular interactions support potential involvement of Hg(II) ions in the Aβ amyloid aggregation processes associated with AD pathology.

## Figures and Tables

**Figure 1 biomolecules-10-00044-f001:**
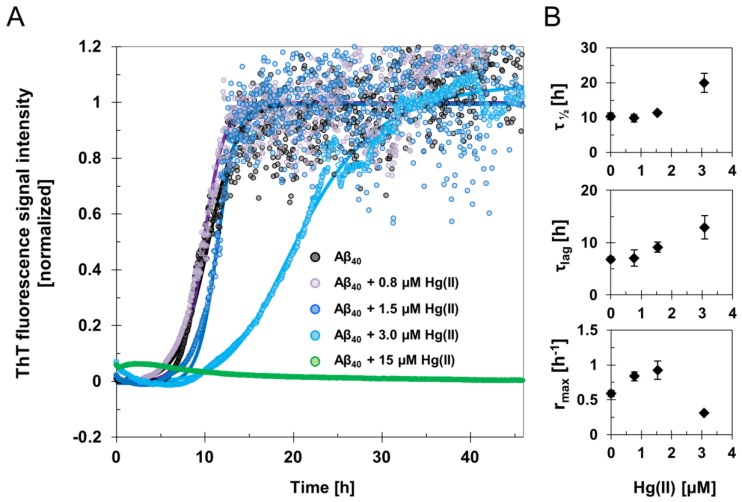
Amyloid fibril formation monitored by ThT aggregation kinetics assay. (**A**) ThT fluorescence signal intensity traces of averaged and normalized data from samples with 15 μM Aβ_40_ peptides in 20 mM sodium phosphate buffer pH 7.4 incubated in the absence and presence of 0.8–15 μM Hg(II) ions at +37 °C under quiescent conditions. The average of four replicates is presented as circles, and the average fit is shown as a solid line for each Hg(II) ion concentration. (**B**) Phenomenological parameters were extracted from sigmoidal curve fitting of the experimental ThT data in (**A**) using Equation (1), yielding the aggregation halftime τ_½_, lag time τ_lag_, and maximum growth rate r_max_. The error bars in (**B**) represent the standard deviation of four replicates.

**Figure 2 biomolecules-10-00044-f002:**
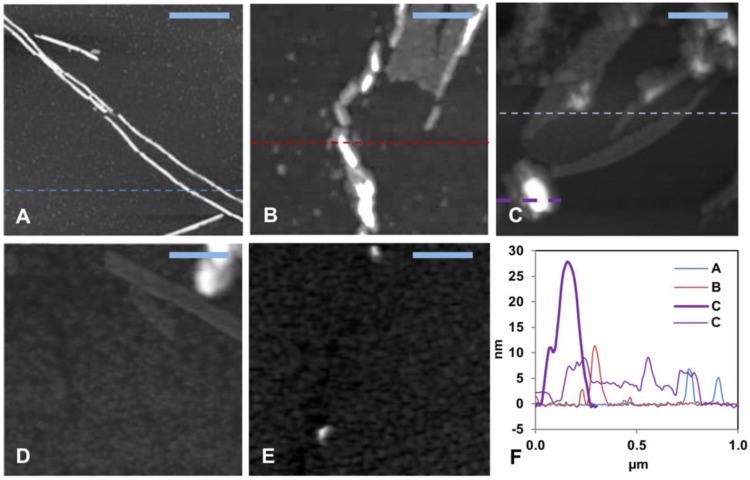
Solid-state AFM imaging of Aβ_40_ fibril formation and morphology. (**A**–**D**) Topographical images of samples taken from the end of the ThT aggregation experiment (~45 h) in Figure 1. (**A**) shows 15 μM aggregated Aβ_40_ peptides alone, (**B**) Aβ_40_ + 0.8 μM Hg(II) ions, (**C**) Aβ_40_ + 1.5 μM Hg(II) ions, (**D**) Aβ_40_ + 3.0 μM Hg(II) ions, (**E**) Aβ_40_ + 15 μM Hg(II) ions, and (**F**) shows cross sectional height information from the images in (**A**–**C**). The scale bars represent 0.25 μm.

**Figure 3 biomolecules-10-00044-f003:**
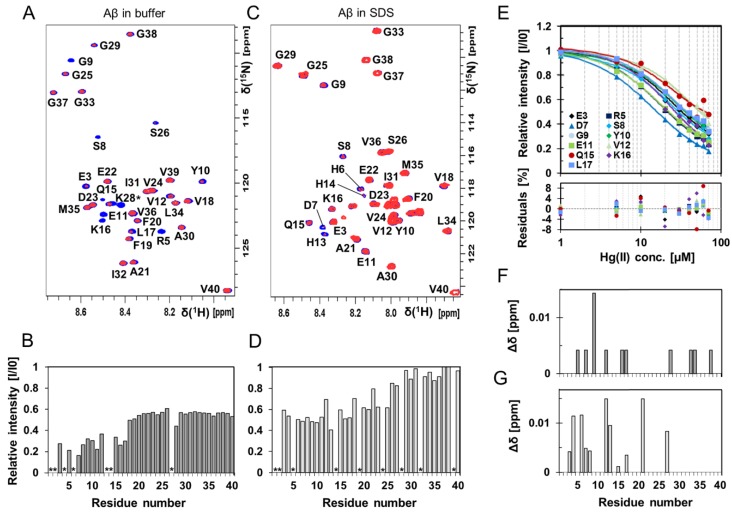
Two-dimensional (2D) NMR ^1^H–^15^N-HSQC experiments showing Aβ_40_ residue-specific perturbations from Hg(II) ions both in buffer and in the presence of SDS micelles. (**A**) 700 MHz ^1^H–^15^N-HSQC spectra of 84 μM monomeric ^13^C–^15^N-labeled Aβ_40_ peptides alone (blue) and in the presence of 80 μM Hg(II) ions (red) in 20 mM sodium phosphate buffer pH 7.35 at +5 °C. In (**B**), relative signal intensities determined from the amplitude of the amide crosspeaks in the two spectra in (**A**) are shown. (**C**) 500 MHz ^1^H–^15^N-HSQC spectra of 84 μM monomeric ^15^N-labeled Aβ_40_ peptides (blue) with 30 μM Hg(II) ions (red) in 20 mM sodium phosphate buffer pH 7.35 and 50 mM SDS at +25 °C, and corresponding relative intensities are shown in (**D**). Residues assigned with a * are not accurately determined or observed because of too fast an exchange with the solvent or due to spectral overlap. (**E**) Relative intensities from ^1^H-^15^N-HSQC spectra (for Aβ_40_ in buffer solution) for selected residues in the N-terminal part of the Aβ_40_ peptide were plotted against Hg(II) concentration and fitted globally using Equation (2), assuming one binding site without any buffer correction. Spectra from 84 μM monomeric ^13^C–^15^N-labeled Aβ_40_ peptides were used for the non-quantitative apparent dissociation constant (K_D_^app*^) determination. The estimated apparent dissociation constant was determined to 11 ± 4 μM. Combined chemical shift differences from the spectra in (**A**,**B**) are shown in (**F**,**G**), respectively.

**Figure 4 biomolecules-10-00044-f004:**
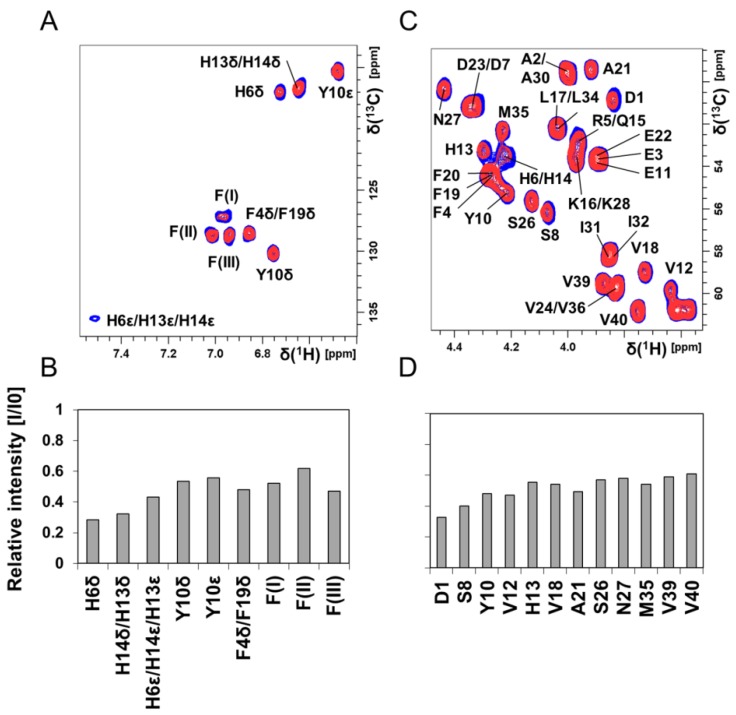
Two-dimensional (2D) NMR ^1^H–^13^C-HSQC experiments showing Aβ_40_ residue-specific perturbations from Hg(II) ions in buffer; 700 MHz NMR data of 84 μM monomeric ^13^C-^15^N-labeled Aβ_40_ peptides in 20 mM sodium phosphate buffer pH 7.35 at +5 °C are shown in (**A**–**D**). (**A**) ^1^H–^13^C-HSQC spectra showing the aromatic region of Aβ peptides (blue) and with 80 μM Hg(II) ions (red). In (**B**), the relative intensities from the spectra in (**A**) are shown. Crosspeaks marked with F(I)–F(III) are resonances from phenylalanine residues, without any detailed assignment. (**C**) ^1^H–^13^C-HSQC spectra showing the C_α_–H region of Aβ peptides (blue) and with 80 μM Hg(II) ions (red). In (**D**), the relative intensities from the spectra in (**C**) are shown. Significant chemical shift changes were not observed. In (**D**), only crosspeaks without any spectral overlap were included in the evaluation.

**Figure 5 biomolecules-10-00044-f005:**
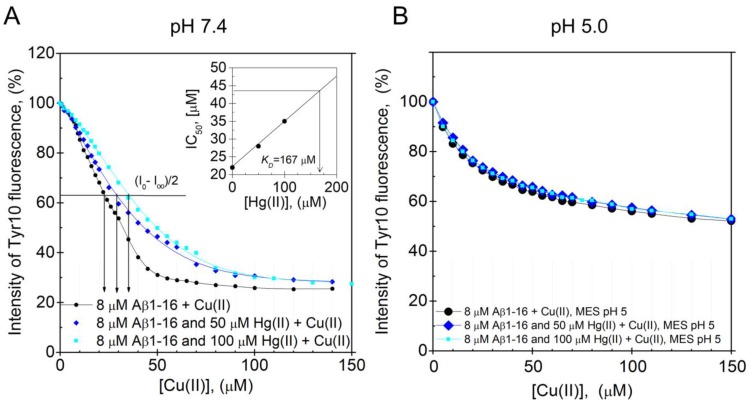
Dissociation constants for the Aβ·Hg(II) ion complex determined from intrinsic fluorescence quenching. Competitive binding using Tyr10 intrinsic fluorescence quenching experiments were performed by titrating Cu(II) ions onto 8 μM monomeric Aβ_16_ peptides in 100 mM NaCl + 20 mM HEPES buffer, pH 7.4 (**A**) or 20 mM MES buffer, pH 5.0 (**B**) at +25 °C in the absence and presence of 50 and 100 μM Hg(II) ions. A similar binding site for Cu(II) and Hg(II) ions was assumed. In (**A**), the apparent dissociation constant for the Aβ_16_·Cu(II) complex was determined to be approximately 22 μM. The relatively high apparent dissociation constant (low affinity) compared to previously reported values [74,75] can possibly be explained by different experimental conditions. The inserted graph in (**A**) shows the IC_50_ for Cu(II) as a function of Hg(II) ion concentration. Extrapolation to the point where the IC_50_ for Cu(II) is increased by a factor of 2 gives an estimated value of 170 μM Hg(II) ions, corresponding to the binding constant. At pH 5.0, the Cu(II) titration series had similar appearances in the absence and presence of Hg(II) ions.

**Table 1 biomolecules-10-00044-t001:** Kinetic parameters of Aβ_40_ fibril formation. ThT fluorescence data reflecting Aβ amyloid formation were recorded in the presence of various concentrations of Hg(II) ions. Aggregation halftimes (τ_½_), lag time (τ_lag_), maximum growth rates (r_max_), and ThT end-point fluorescence amplitudes were derived from sigmoidal curve-fitting to Equation (1) and are also presented in Figure 1.

	τ_½_ [h]	τ_lag_ [h]	r_max_ [h^−1^]	ThT End-Point [a.u]
15 μM Aβ_40_	10.4 ± 0.9	6.8 ± 0.4	0.6 ± 0.04	7800 ± 800
15 μM Aβ_40_ + 0.8 μM Hg(II)	9.8 ± 1.1	7.1 ± 1.6	0.8 ± 0.06	6000 ± 2000
15 μM Aβ_40_ +1.5 μM Hg(II)	11.4 ± 0.5	9.1 ± 1.0	0.9 ± 0.13	5500 ± 300
15 μM Aβ_40_ +3 μM Hg(II)	19.9 ± 2.7	12.9 ± 2.2	0.3 ± 0.01	3500 ± 1100
15 μM Aβ_40_ +15 μM Hg(II) *	n/a *	n/a *	n/a *	n/a *

* n/a—not applicable: the kinetic traces for Aβ_40_ in the presence of 15 μM Hg(II) ions could not be described by sigmoidal curve fitting.

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
