# Peer review of "Mercury and Alzheimer’s Disease: Hg(II) Ions Display Specific Binding to the Amyloid-β Peptide and Hinder Its Fibrillization"

_biomolecules, 2019, doi:10.3390/biom10010044_

Round 1
Reviewer 1 Report
The manuscript clearly describes the binding of Hg(II) ions to Aβ and its role in fibrillization. The MS is well written in general, and relevant information is included. Below there are some specific comments and suggestions that should be attended before accepting this MS for publication in this journal.
Please add information about Hg and blood-brain barrier interactions. Hg generally binds to (-SH) and –(SeH) group. In Aβ, which group is involved in binding with Hg and what is the type of bond. Is the role of Hg in AD and Hg-intoxication are same? Discuss.
Reviewer 2 Report
In the paper “Hg(II) ions display specific binding to the amyloid-βpeptide and impede its fibrillization”, Wallin and co-workers have studied the in vitro interactions between Aβ peptides and Hg(II) ions using complementary biophysical techniques. They have also investigated the effect of Hg (II) ions in the in vitro amyloid aggregation of the Aβ-peptide.
Although the biological relevance of Hg(II) binding to the Aβ-peptide seems to be very low (low intracellular Hg concentration, very low binding affinity between Hg (II) and Aβ-peptide), this study contributes to the understanding of the molecular mechanisms underlying metal binding in amyloid aggregation. This study is nicely performed, and the conclusions of the Authors are fully supported by the results.
The paper is generally well written, and the goal of the research is clear. For the above reasons, in my opinion it is suitable for publication in “Biomolecules”.
Reviewer 3 Report
The aim of this work was to investigate the effect of Hg over Ab fibrillization. Although this type of pure in vitro studies have low clinical implications, they are worth to be published since they can set the basis for further research with more applicability potential. Importantly, here the authors investigate the always controversial relationship between amyloids and metals.
My major concern is from a methodological point of view. I see there are different experimental conditions that I find hard to be justified. My main comments are:
- Most of the work is done with Ab40, why then did the authors include Ab42, which is tested in totally different conditions (higher temperature, different format, stirring vs. quiescent, different Hg salt, different Hg:Ab molar ratio, etc…). Considering that Ab42 is intrinsically more prone to aggregate than Ab40, it is impossible to obtain any conclusion further than the message that Hg also affects Ab42. In this case, I strongly recommend sending Figure 2 to Supplementary material.
- Figure 2 (which I recommend not to include in the main manuscript) does not contain any dispersion measure. Please include either the individual dots for each replicate or error bars.
- Figure 1 and Table 1 show the results of the 96-well plate Ab40 aggregation. I understand that AFM (Fig 3) is done with the species collected from those aggregation reactions. But there is a total mess with the Hg concentrations used, with different values between Figure 1A (1.5, 3, 15 uM), Figure 1B panels and Table 1(0.2, 0.8, 1.5, 3 uM ???), Figure 1 caption (0.2-15uM) and Figure 3 (0.8, 1.5, 3, 15uM). Please check this; the experiments shown should be the same in all panels.
- When reading the work, one has the idea that high Hg concentration inhibits amyloid formation (understood as low ThT signal in the kinetics and nothing in the AFM images). However, from NMR experiments, the authors state “indicates that Hg(II) ions promote formation of large Aβ aggregates, where some of them are too large to be observed with HSQC NMR and some simply precipitate out of the solution”. In the conclusion lines 416-419, they state that at equimolar concentration, Hg promotes non-fibrillar aggregates. I see the line of argument to explain that ThT signal decrease in the presence of Hg is due to formation of non-amyloid aggregates (they might not bind ThT). However, what I cannot understand is why these big non-fibrillar aggregates do not appear in AFM images (they should be visible in Figure 3E; however, this photo shows nothing). Please discuss how these data fit together; I do not manage to see any reasonable explanation.
- I suggest including Figure Supp 5 in the main manuscript, as it does not overlap with any other figure and indeed it is the support of the whole results section 3.4.
- Regarding this binding affinity experiment, in lines 356-359, the authors explain that by extrapolation, the apparent Kd for the Hg complex is about 170uM. I admit this could be just my limited knowledge on this issue, but why was this concentration not experimentally tested?
- Please check first sentence of the introduction section
- Line 125: “A similar approach was used to follow Ab42 fibrillization…” I don’t think it’s a similar approach as all conditions are totally different from Ab40 kinetics.
- Line 421-422: “The Hg(II) ions bind also to Aβ peptides positioned in SDS micelles (Figures 4C,D,G), indicating that Hg(II) ions can bind Aβ peptides located in cellular membranes.” That is an over-interpretation of the data. If you want to consider SDS as a model of cellular membranes (totally debatable) , at least substitute “indicating that Hg(II) ions can bind Aβ peptides...” by “SUGGESTING that Hg(II) ions COULD bind Aβ peptides...”.
Reviewer 4 Report
This is an excellent work that investigates the effect of Hg(II) on amyloid beta aggregation. It is an interesting and important work. It is expected to be a high impact to the community. The authors apply extensive experimental techniques to investigate the effect of Hg(II) on Abeta fibrillation. The paper is well-written and it is strongly recommended for publication in Biomolecules after minor issues to address.
Minor points:
The authors demonstrate that the Hg(II) ions inhibit amyloid beta fibrillation both for Abeta(1-40) and Abeta(1-42). The inhibitory function is strongly exhibited in the ratio Hg(II):Abeta of 1:1. The authors claim along the ms that the inhibitory effect of Hg(II) is concentration-depended. This is not what is claimed in the title of the paper. It is recommended to change the title accordingly. It is not surprising that Hg(II) binds to the N-term of abeta. This was shown for other transition metal ions. It is not clear from the ms, how the authors suggest from the experiments that performed the specific amino acids that bind to Hg(II). The authors should provide a discussion with regards this issue and compare to other metal ion transitions: (1) Miller, Y., Ma, B. and Nussinov, R. 2012. Metal binding sites in amyloid oligomers: complexes and mechanisms. Coord. Chem. Rev., 256: 2245-2252. (2) Wineman-Fisher, Bloch D. N. and Miller Y. 2016 The challenges in studying the structures of metal- amyloid oligomers related to Type 2 diabetes, Parkinson's disease and Alzheimer's disease. Coord. Chem. Rev., 327, 20-26 Can the authors provide an insight whether Hg(II) interacts with abeta, intramolecularly (i.e. each Hg(II) ion binds one abeta peptide) when the ratio is 1:1, and Hg(II) interacts with abeta intermolecularly (i.e. each Hg(II) ion binds two abeta peptide) when the ratio is 1:2. This is an interesting issue that the authors should discuss in the paper.Author Response
Please see the attachment.

Round 2
Reviewer 3 Report
I still believe that including the Ab42 experiment in the paper is comparing apples and oranges. Even the kinetic platform has nothing to do (quartz cuvettes vs 96-well plate). That is why I think it should be shifted to supplementary.
But if the authors keep it in the main manuscript, at least it should be scientifically and statistically sound. When in the previous report I commented on adding "dispersion measure (individual dots for each replicate or error bars)" I assumed it was performed more than once.
If so, I do not agree with authors in showing only one experiment per condition, or without error bars. There are no reasons for that (while they did a very elegant job with Ab40 experiment, for which the number of replicates is stated and dispersion measures are shown).
If Fig 2 is part of the main manuscript, it should be clearly shown how many replicates/times was the experiment repeated and the associated dispersion. This is the minimum of scientific standards.
Showing or doing n=1 per condition is not of quality enough for a journal like Biomolecules.
Author Response
We have now moved Figure 2 to the supplementary material. We also moved supplementary figure S5 to the main manuscript, as reviewer #3 had suggested earlier. With these changes we hope that reviewer #3 and everybody else will be happy with the current version of the manuscript.